# Megacity Wastewater Poured into A Nearby Basin: Looking for Sustainable Scenarios in A Case Study

**Silvia Chamizo-Checa** [1,2], **Elena Otazo-Sánchez** [2,*] , **Alberto Gordillo-Martínez** [2],
**Juan Suárez-Sánchez** [1], **César González-Ramírez** [2] and **Hipólito Muñoz-Nava** [1]

1    Facultad de Agrobiología, Universidad Autónoma de Tlaxcala, Autopista Tlaxcala-San Martin Texmelucan
     km 10.5, CP 90120 Tlaxcala, Mexico, Mexico; silchamiz@gmail.com (S.C.-C.);
     jsuarezs71@hotmail.com (J.S.-S.); hipolito78@hotmail.com (H.M.-N.)
2    Area Académica de Química, Universidad Autónoma del Estado de Hidalgo, Carretera Pachuca-Tulancingo
     km. 4.5, Mineral de la Reforma, CP 42184 Pachuca, Hidalgo, Mexico; gordillo@uaeh.edu.mx (A.G.-M.);
     ccr_ramirez@yahoo.com (C.G.-R.)
*    Correspondence: elenamariaotazo@gmail.com; Tel.: +52-7717172000 (ext. 2208)

**Abstract:** Megacity sewage creates socioeconomic dependence related to water availability in nearby areas, especially in countries with hydric stress. The present article studies the past, current, and future water balance progression of realistic scenarios from 2005 to 2050 in the Mezquital Valley, the receptor of Mexico City untreated sewage since 1886, allowing for agriculture irrigation under unsustainable conditions. The Water Evaluation and Planning System (WEAP) was used to estimate water demand and supply, and validation was performed by comparing results with outflow data from the Tula River. Simulated scenarios were (1st) steady-state based on inertial growth rates (2nd) transient scenario concerning the influence of forecasted climate change perturbations in surface water and hydric stress for 2050; and (3rd) the previous scenario appending scheduled actions, such as 36% reduction in imported wastewater and the startup of a massive Wastewater Treatment Plant, allowing for drip and sprinkler irrigation from the year 2030. The main results are as follows: (a) in the period 2005–2017, 59% of the agriculture depended on flood irrigation with megacity sewage; (b) the outcomes of water balance scenarios up to 2050 are presented, with disaggregated sectorial supply of ground and superficial water; (c) drip irrigation would reduce agriculture demands by 42% but still does not guarantee the downflow hydroelectric requirements, aggravated by the lack of wastewater supply from 2030. This research highlights how present policies compromise future Valley demands.

**Keywords:** water demand; megacity wastewater; hydrological balance scenarios

## 1. Introduction

Water scarcity is the most prominent environmental challenge the world faces nowadays. The trends indicate that by 2025 two-thirds of the world's population will experience severe water shortages [1]. Megacities are considered Global Risk Areas and a sink of natural resources. Their sewage drains in huge volumes whose sustainable use mitigates water scarceness in the neighboring areas [2]. In the Middle East, Asian, and African regions with water scarcity, megacities are boosting; the agriculture sector which increases water irrigation demand and water resource planning is a challenge to guarantee economic and social development [3]. Freshwater demands exceed 15–20% of the supplies and the 2025 trends predict severe water shortages for over half of the world's population [3]. On a basin scale, water management needs efficient multidisciplinary integration [4].

Megacity water supply and management has been a research interest of policy scientists in recent years because of the challenge of securing water and sanitation services. In emerging countries, cities' growth rates are still increasing, and water management gets worse [5,6]. However, the impact of large cities on their proximity areas has garnered less attention. Megacity sewage is commonly used without treatment for agriculture irrigation despite the health risks and environmental damage [7,8]. Although it raises organic carbon and nutrients for improving crops, wastewater irrigation promotes serious contamination risks to the groundwater and soil [9]. The World Health Organization [10] warns on the high concentration of pathogens and chemicals in soils; still, it remains a widespread practice in developing countries [11]. Water reuse applications have been accepted solutions with substantial challenges concerning the selection of water treatment technologies [12].

### 1.1. Hydrological Balance Model

Hydrological models are useful tools for calculating water balances in case study scenarios. They are widely employed to understand and predict the local water cycle (occurrence, circulation, and distribution) and the future behavior of demand/supply issues based on realistic or hypothetical conditions as proposal adaptation actions. The results are relevant to water management and planning [13], and the model selection depends on the mathematical approaches, the available data, and the appropriate system representation. A comprehensive review of the most relevant hydrological models and features was published recently [14].

Climate change adds uncertainty to the modeling results due to lowering precipitation and rising temperatures, intensified in semiarid regions [15]. Precipitation is the primary supplier of the water cycle and the most affected by global change, causing differences in rain intensity, floods, and droughts [4]. Rain infiltration and surface flows contribute to evapotranspiration output, whose proper calculation is essential to an accurate hydrological balance [3].

The Water Evaluation and Planning (WEAP) System is one of the most versatile predictive models for analyzing federal programs related to water scenarios. It integrates technical parameters such as demand, supply, infiltration, runoff, crop requirements, flows, storage, discharges, and ecological issues [16], and the results allow analysis of political actions. The web page displays plenty of papers and reports uploaded from the world community members [17]. That is why WEAP is suitable for the modelling of complex basins, such as those altered by the proximity to megacities, as is the case of the Mezquital Valley, which is a semidesertic area threatened by climate warming and the high growth of its industry and agriculture.

### 1.2. Study Case

With a population of 20.8 million people [18], Mexico City is the 11th global biggest megacity in the world. It is located in the Mexican Central Plateau, whose water infrastructure supply and sewage discharge systems were described in detail by [19,20]. For over 150 years, the untreated wastewaters from the megacity have been drained to the semiarid Mezquital Valley, which has become one of the most environmentally impaired regions in the country. It is located 50 km to the north of Mexico City in the southwest area of Hidalgo State, between 19°45′ and 20°40′N and 98°44′ and 99°36′ W at 1910–2150 masl (meters above sea level) (see Figure 1).

Proximity to the megacity promoted the local development of this semidesertic valley that supports agriculture and industry, making it economically dependent on the neighboring metropolis but causing severe environmental damage due to the unmanaged wastewater flood irrigation. The water-dependent valley is at risk of being water-suppressed if the governing board asks for more treated wastewater returns as most cities do.

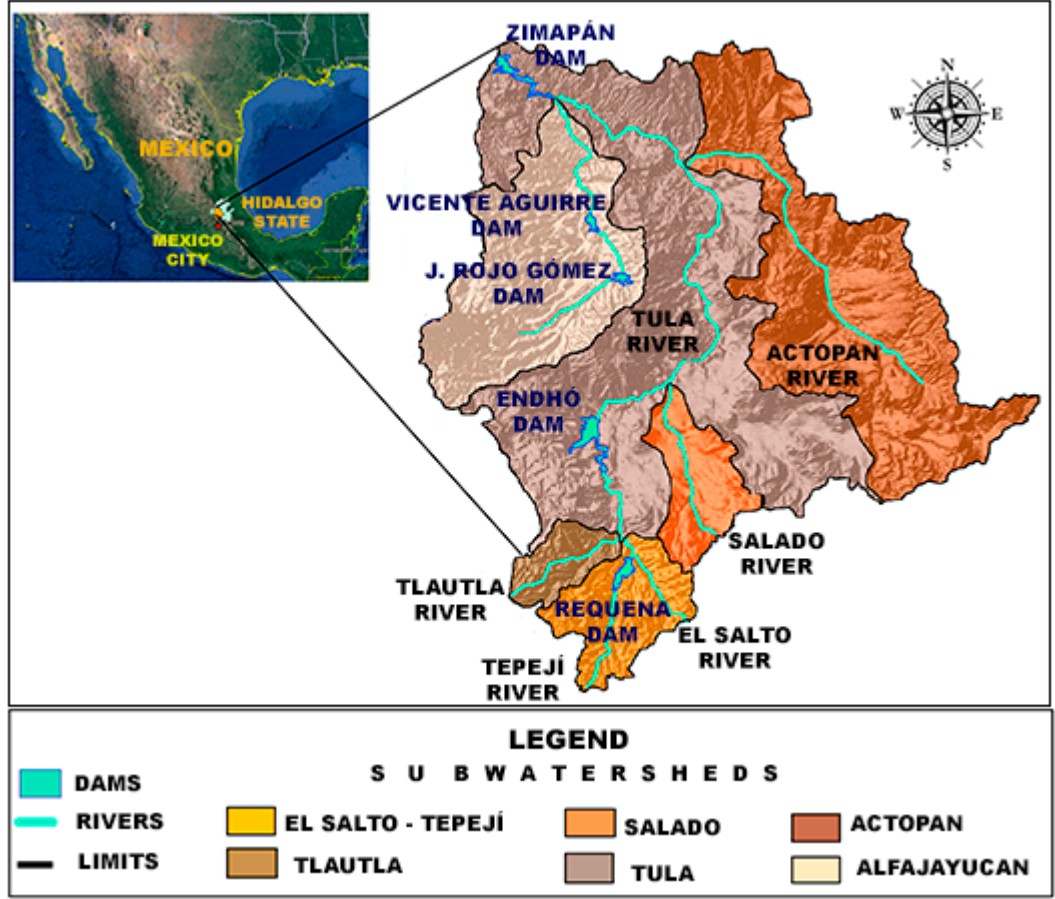

**Figure 1.** Mezquital Valley Basin located in the Central Mexican Plateau. Major rivers, dams, and six sub-basins delimited by the runoff criteria from the elevations map.

In 2005, 711,450 inhabitants lived in 5045 km$^2$ in the valley area. More than half of the population is indigenous and highly marginalized [21]. Untreated wastewater (1663 Mm$^3$) is still drained yearly into the area for agricultural flood irrigation [7].

Figure 2 shows the interaction between the valley, Mexico Megacity, and the primary sewage supply sources. Three main flows convey the wastewater to the valley: (a) The Great Sewage Channel (Gran Canal) (built in the 19th century, improved in 1950, and expanded in 2000), which is affected by subsidence and leads to wastewaters being pumped to the valley; (b) the Deep Sewerage Tunnel (Emisor Profundo) constructed in 1975 (153 km long at 200 m depth) and operating at 200–340 m$^3$/s with significant operational and maintenance problems; and (c) the 62 km underground East Transmitter 7 m diameter tunnel (Emisor Poniente), which became operational in the 60s and was renovated and enlarged in the last 10 years. Additionally, the Tequisquiac and the Nochistongo Tagus tunnels join the El Salado and El Salto Rivers, respectively [22]. A channel was recently constructed to merge El Salado River with El Salto River, allowing for the treatment of the whole wastewater inflow.

The Tula River is the mainstream of the valley, receiving the megacity wastewaters from the effluents El Salto and El Salado. The main rivers collect the rain runoff and local sewage from other secondary streams, supplying the water to the irrigation districts of the valley, and finally, the basin outflow heads towards the Zimapan hydroelectric dam [23]. El Salado River crosses the Mezquital Valley and merges with El Salto in the Requena Dam, creating the Tula River, which pours into Endho, the most significant Mexican wastewater reservoir (200 Mm$^3$). It acts as a regulating/supply vase for the agriculture irrigation districts Tula, Ajacuba, and Alfajayucan through channel networks [24].

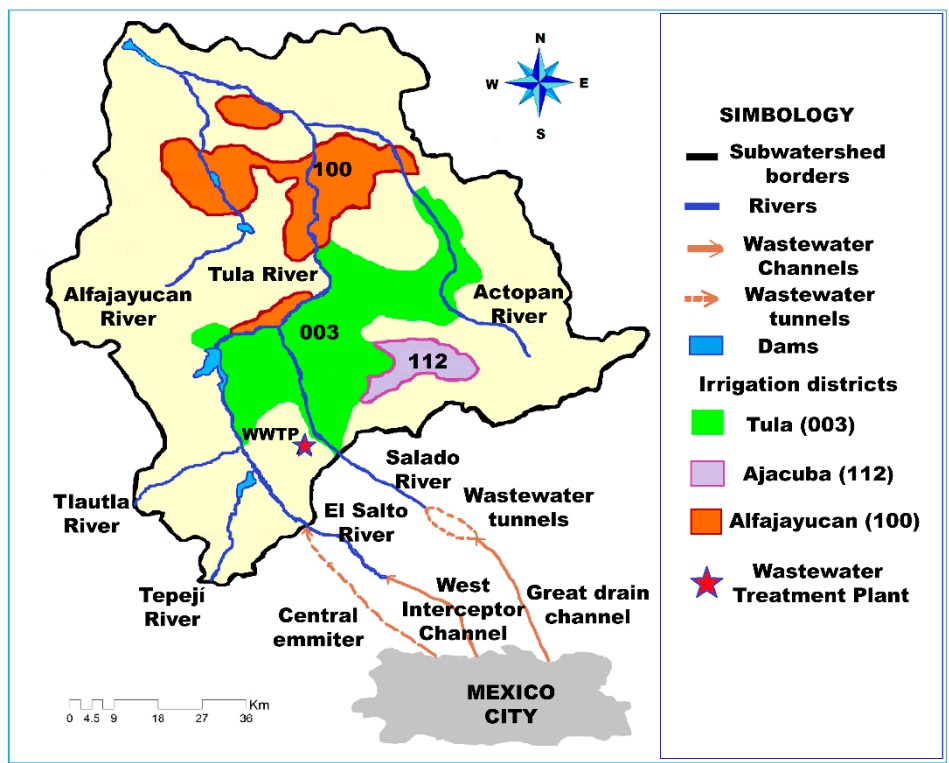

**Figure 2.** Schematic representation of the interaction between the Mezquital Valley and Mexico Megacity and the location of the irrigation districts in orange, green, and grey. The Central Emitter, West Interceptor, and Great Canal are the wastewater entrance. Note that Actopan and Alfajayucan Rivers are not linked with the Megacity, but receive the local wastewaters, poured without treatment. Adapted from free online resource [24].

The irrigation permeates the soil and modifies the natural groundwater recharge. The irrigation rates go from 1500 to 2200 mm/y depending on crop and soil type [25]. Agricultural consumption reduces the final output, which flows to the Zimapan Hydroelectric dam, affecting its generation capacity.

There are many reports on the Mezquital Valley environmental damage, such as water contamination [26–30], agriculture emissions [25], health risks [7,31], soil degradation [32] and pollution [33–35], and unsustainable agricultural practices [36–39], among others.

Recently, the Federal government completed the construction of a large wastewater treatment plant (WWTP) in Atotonilco de Tula, just at the entrance of the Valley, in the sewage input. Its lowest capacity is 23 $m^3$/s for the dry season, with an additional 12 $m^3$/s unit for rainy times. The WWTP started operation at 30% of its capacity in 2017 (300 $m^3$/y) and 100% since January 2020 [22,40,41]. The water balance studies in the Mezquital Valley have focused on groundwater, with unusual shallow aquifer replenishment due to unmanaged recharge [24,42]. There are no reports about the internal fluxes and interactions in the water cycle, as well as prospective analysis of water use.

The purposes of this work are (i) to quantify how much agriculture depends on the unsustainable management of the sewage received from Mexico Megacity due to the lack of suitable policies, (ii) to achieve the hydrological balance of the case study, considering the basin complexity in terms of clean and wastewater, and the sectorial water allocation (industries, agriculture irrigation, population, services, internal returns, and final outflow that supplies a hydroelectric dam downstream), and (iii) to provide in-depth insight about the past (2005), recent (2017), and future (2030 and 2050) balance scenarios which consider local climate changes and planned mitigation actions, such as the startup of the giant Wastewater Treatment Plant in 2017 that allows for sustainable irrigation alternatives that will progressively take over the unsustainable flow irrigation practices until the year 2050.

This work will contribute to the understanding of the megacities assessment towards their immediate environment concerning wastewater management, whose impact is not equal in all countries nor all cities. Its findings are the main contribution of this work.

## 2. Materials and Methods

### 2.1. Model Description

WEAP is considered a computational laboratory for examining water management strategies in physical systems. The natural and technical components are represented in a network-like scheme with interconnected model elements. The WEAP model allows the platform to be driven by user-defined priorities, preferences, and environmental requirements for the various nodes. The water allocation problem is solved using linear programming on a daily or monthly basis. For estimating evapotranspiration, runoff, interflow, baseflow, and percolation the model uses empirical equations [43].

### 2.1.1. Schematic Model

This section consists of drawing the model diagram (Figure 3), representing the mainstream Tula River in a south–north flow direction, secondary streams, water importations, reservoirs, demands, and their interactions [16].

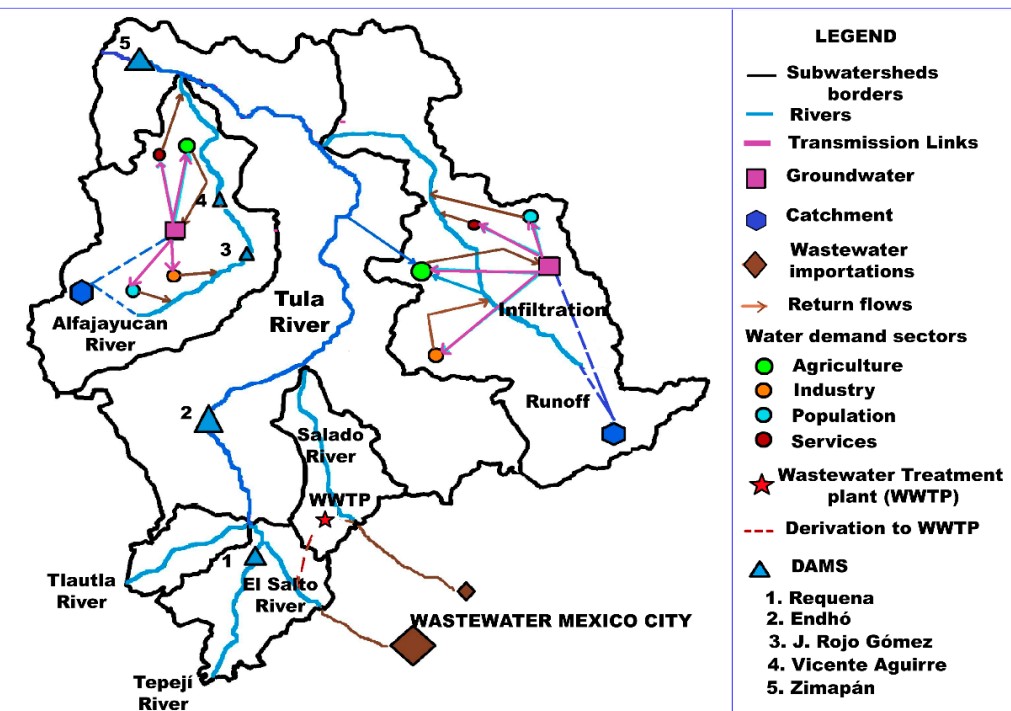

**Figure 3.** Conceptual model of the Mezquital Valley Basin.

### 2.1.2. Assumptions for the Baseline Scenario (2005)

- The valley is considered a basin that compiles a set of six adjacent sub-basins. Figure 1 shows the elevation map of the area, which delimits sub-basins. The GIS layer depicts six sub-basins: three small ones in the southern region (Tlautla, Tepeji-El Salto, and El Salado Rivers) and three bigger ones in the center and north (Actopan, Alfajayucan, and Tula Rivers).
- The superficial wastewater importation from Mexico City enters in the southern area through the El Salto and El Salado Rivers. Both flows merge as a one-point entrance by the connecting channel between them.

- The startup of WWTP in Atotonilco de Tula occurred in 2017 at a 30% capacity and reached 100% as of 2020. It treats about 60% of the imported sewage at its maximum capacity.
- There are shallow and deep aquifers. The last ones are not contaminated by the internal wastewater returns from the surface (rivers, channels, and irrigation) as reported by [24,29]. Sewage infiltrates the shallow aquifers that supply agriculture demands in areas outside the irrigation districts.
- The model considers six aquifers based on the orography, which determines the catchment areas. It does not assume the political division criteria reported by [24].
- The value of consumptive water for each irrigation district was distributed and divided by the area, based on the official reports [22]. The consumptive water use for individual crops was considered similarly because the flood irrigation distributes water proportionally to the sown area, regardless of the type of crop.
- Starting from 2020, 31% of the wastewater importation from Mexico City is to be allocated in Mexico State. Scenario 3 considers this sharp reduction as a realistic transient scenario. The program provides the mathematical expressions which best fit the conceptual model assumptions and aims.

### 2.1.3. Data Model

This module required the following input information: water supply and demand sites for surface and groundwater (domestic, agriculture, industrial, and service sectors) [44], cover vegetation, land use [45], crop coefficient (Kc) [46,47], monthly precipitation, and temperature [48]. Evapotranspiration (Eto), runoff, and infiltration were calculated to simulate the hydrological process by the "rainfall-runoff" module in the simulation platform [49]. Tables 1–4 show the input data used for the calculation with the mathematical tools of WEAP.

**Table 1.** Water bodies, urban and vegetation cover areas (%) in the Mezquital Valley.

| Sub-Watersheds Rivers | Area (km$^2$) | % | | | | | |
|---|---|---|---|---|---|---|---|
| | | Water Bodies | Urban | Temporal Agriculture | Irrigation Agriculture | Pasture Land | Forest and Scrub |
| Tlautla | 172 | 0.0 | 9.0 | 14 | 23 | 33 | 21 |
| El Salto-Tepejí | 260 | 1.15 | 8.5 | 13 | 23 | 33.5 | 21 |
| Salado | 275 | 0.0 | 21 | 38 | 15 | 5.5 | 20.5 |
| Alfajayucan | 850 | 0.8 | 1.2 | 12 | 22 | 25 | 39 |
| Actopan | 1320 | 0.0 | 7 | 11 | 27 | 8.0 | 47 |
| Tula | 2168 | 1.0 | 16 | 29 | 20 | 12 | 21 |
| Mezquital Valley Basin | 5045 | 0.1% | 10.7% | 42.6% | | 17.6% | 29% |

**Table 2.** Parameters used to calculate the infiltration coefficients Kfc and Kp.

| River Sub-Basin | Texture Soil | | | Kfc | Slope (%) | Kp |
|---|---|---|---|---|---|---|
| | Sandy | Silty | Clay | | | |
| El Salto-Tepejí | 35 | 60 | 5 | 0.315 | 22.3 | 0.15 |
| Tlautla | 28 | 68 | 4 | 0.33 | 18.66 | 0.2 |
| Salado | 30 | 67 | 3 | 0.33 | 10.02 | 0.2 |
| Actopan | 35 | 62 | 3 | 0.32 | 16.57 | 0.2 |
| Alfajayucan | 45 | 53 | 2 | 0.30 | 14.07 | 0.2 |
| Tula | 32 | 66 | 2 | 0.33 | 18.75 | 0.15 |

**Table 3.** Parameters used for Kv determination.

| River Sub-Basin | Urban Areas | Agriculture | Pasture Land | Forest and Scrub | Water Bodies | Kv |
|---|---|---|---|---|---|---|
| El Salto-Tepejí | 8.50 | 36.00 | 33.50 | 21 | 1.15 | 0.12 |
| Tlautla | 9.00 | 37.00 | 33.00 | 21 | 0.00 | 0.12 |
| Salado | 21.00 | 53.00 | 5.50 | 21 | 0.00 | 0.09 |
| Actopan | 7.00 | 38.00 | 8.00 | 47 | 0.00 | 0.11 |
| Alfajayucan | 1.20 | 34.00 | 25.00 | 39 | 0.80 | 0.11 |
| Tula | 16.00 | 49.00 | 12.00 | 21 | 1.00 | 0.10 |

**Table 4.** Ci and Ce values of Mezquital Valley sub-basins.

| River Sub-Basin | Kfc | Kp | Kv | Ci | Ce |
|---|---|---|---|---|---|
| El Salto-Tepejí | 0.315 | 0.15 | 0.12 | 0.59 | 0.42 |
| Tlautla | 0.33 | 0.2 | 0.12 | 0.65 | 0.35 |
| Salado | 0.33 | 0.2 | 0.09 | 0.62 | 0.38 |
| Actopan | 0.32 | 0.2 | 0.11 | 0.63 | 0.37 |
| Alfajayucan | 0.30 | 0.2 | 0.11 | 0.61 | 0.39 |
| Tula | 0.33 | 0.15 | 0.10 | 0.58 | 0.42 |

Water Supply Sources. The groundwater recharge considered rainwater infiltration and irrigation water returns. The total surface water included rain runoff, imports from the megacity, and domestic wastewater generation.

Cover Vegetation. The vector shapes of land use information [45,50] were classified and quantified using Quantum GIS software. Table 1 shows the data of urban sprawl, water bodies, and vegetation areas in each sub-basin and the whole Mezquital Valley. The agriculture areas are predominant (42.6%), followed by forest (29%), pastureland (17.6%), and urban areas (10.7%).

Runoff and infiltration coefficients estimation. Equation (1) calculates the infiltration coefficients (Ci) [51].

$$Ci = (Kp + Kv + Kfc) \tag{1}$$

where Kp, Kfc, and Kv are the infiltration fractions corresponding to the slope, soil type, and vegetation cover, respectively (see Tables 1–3).

The runoff coefficient (Ce) was calculated by Equation (2) [52] (see Table 4).

$$Ce = 1.0 - Ci, \tag{2}$$

Climate data: precipitation, temperature, and evapotranspiration (ETo). Eleven meteorological stations located in the valley provided the climate information through ERIC Fast Data Extraction software [53]. Low precipitation occurred from May to October (450–550 mm/y) while mean temperatures ranged between 15 and 19 °C.

Evapotranspiration (ETo) was calculated monthly in the base year (2005) using the standardized Penman–Monteith method [54] (Equation (3)), suitable for the daily and monthly estimation [55]

$$ETo = \frac{0.408\Delta(R_n - G) + \gamma \frac{900}{T+273} u_2(e_s - e_a)}{\Delta + \gamma(1 + 0.34u_2)} \tag{3}$$

Equation (3) includes parameters related to the energy exchange. $R_n$ is the net radiation flux density on the crop surface (MJ m$^{-2}$ d$^{-1}$), $G$ is the soil heat flux density (MJ m$^{-2}$ d$^{-1}$), and $\Delta$ is the slope of vapor pressure–temperature curve (kPa °C$^{-1}$). $T$ is the air temperature (°C), ($e_s - e_a$) is known as vapor pressure deficit, because $e_a$ and $e_s$ are the actual vapor pressure (kPa) at dewpoint and air temperature, respectively. The wind speed at 2 m high is $u_2$ (m s$^{-1}$) and $\gamma$ is the psychrometric constant (kPa °C$^{-1}$) [56].

Mezquital Valley Crops. The main crops are maize (*Zea mays*), alfalfa (*Medicago sativa* ), bean (*Phaseolus vulgaris*), barley (*Hordeum vulgare*), wheat (*Triticum aestivum*), and oat (*Avena sativa*) [57]. The crop coefficient (Kc parameter) considers the harvest period for each crop whose maximal yield values were calculated in kg/ha, considering the growing periods and water requirements [58] for oat [59], wheat [60], bean [61], barley, and corn [46]. Alfalfa presents the highest yield [62,63]. Table 2 shows monthly Kc values and crop yields [47].

Water Demand. CONAGUA's official website provided the 2005 information of the water demand database [44]. Data was worked up by sectors and divided into groundwater (GW) and surface water (SW) for each sub-basin, producing the input demand data shown in Table 5.

**Table 5.** Groundwater and surface demand in the Mezquital Valley sub-basins for the year 2005.

| River Sub-Basin | Groundwater Demand (Mm³/y) | | | | | Surface Water Demand (Mm³/y) | | |
|---|---|---|---|---|---|---|---|---|
| | Population | Agriculture | Industrial | Services | Total | Agriculture | Industrial | Total |
| El Salto-Tepejí | 6.3 | 2 | 8.4 | 0.4 | 17.1 | 18.4 | 3.3 | 21.7 |
| Tlautla | 1.1 | 0.3 | 1.5 | 0.1 | 3 | 5.5 | 0.8 | 6.3 |
| Salado | 6.9 | 4.6 | 69.5 | 0.8 | 81.8 | 125 | 0.5 | 125.5 |
| Actopan | 13.8 | 20.8 | 2.2 | 0.6 | 37.4 | 175 | 0.1 | 175.1 |
| Alfajayucan | 3 | 1.7 | 0.3 | 0.1 | 5.1 | 135 | 1.6 | 136.6 |
| Tula | 25.7 | 14.8 | 41.4 | 4 | 83 | 630.1 | 20.7 | 645 |
| Total | 56.8 | 44.2 | 123.3 | 6 | 230.3 | 1089 | 27 | 1116 |

Validation was performed by plotting the 2005–2010 period of the calculated Tula River annual output vs. the experimental Tula River outflows data assessed in the Ixmiquilpan Hydrometric Station, near the output of the basin [22] (see Figure 4).

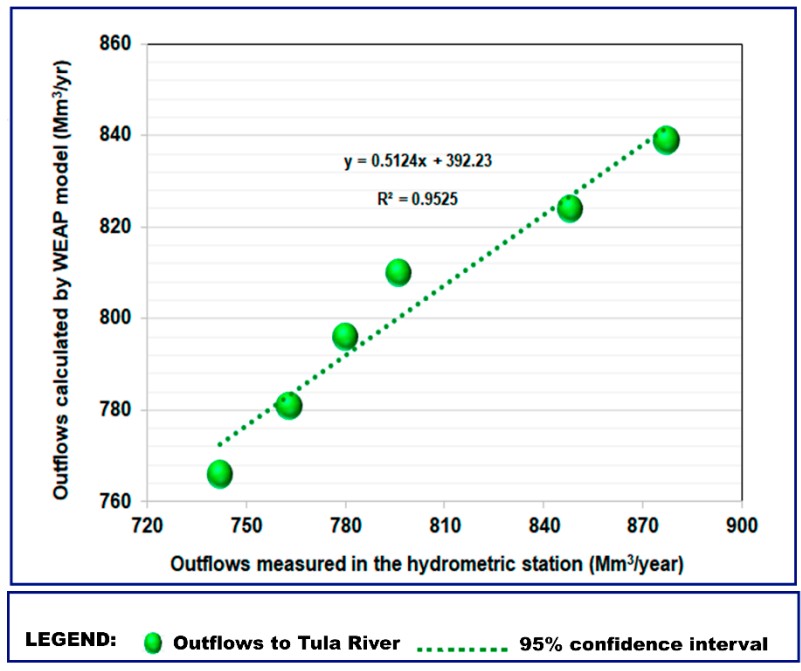

**Figure 4.** Water Evaluation and Planning System (WEAP) model validation. Comparison between model outflows in the Tula River vs. experimental data assessed in the Ixmiquilpan Hydrometric Station in the 2005–2010 period.

*2.2. Scenarios: Steady-State and Transient Conditions*

The hydrological balance scenarios were simulated for years 2017, 2030, and 2050.

(#1) Steady-State Scenario. It simulates the inertial tendency (BAU, business as usual), even though it includes the WWTP startup since 2017. Table 2 shows the population projections data [64]. The industrial growth rates were from the Mexican Business Information System [65]: chemical, 0.45%; textile, 3%; food, 3.7%; metallurgical, 3.7%; and lime and cement, 2.5%. The irrigated areas expansion was assumed to equal the crop annual growth rates (maize, 1.7%; beans, 0.88%; alfalfa, forages, and vegetables, 1.27%) [46].

(#2) Climate Change Scenario. It considers Scenario #1 + climate change perturbation. The average temperature would increase 2 °C from 2005 to 2050 [66], and precipitation would decrease by 6.5% from 2000 to 2030 (0.06% per year) [67].

(#3) Adaptation Scenarios. The perturbations are related to irrigation alternatives and sewage import reduction. It considers Scenario #2 + decreased wastewater imports + each irrigation technology the most likely, with a 36% decrease in wastewater imports since 2020 due to the startup of five sewage treatment plants and the fact that Mexico State will catch 606 Mm$^3$/y. Additionally, the imported surface water will fit the quality requirements for sprinkling and drip irrigation technologies that will be analyzed separately from 2020, instead of the current flood irrigation with water savings of 25% and 43%, respectively [68].

## 3. Results and Discussion

### 3.1. Model Validation

There is a significant linear correlation between the Tula River calculated outflows and the experimental data ($R^2$ = 0.9525) with 95% confidence (Figure 4). Experimental outflow data were the official reported values for each year in the same period (2005–2010) [22]. The outflow values have been slowing down over time due to the rising temperature, lowering precipitation, unsustainable and uncontrolled flood irrigation, and agriculture development.

### 3.2. Model Boundaries Result in the Base Year 2005 (Baseline)

Water supply sources and inflows. Figure 5 shows the primary data for each sub-basin of the Valley. The infiltration values were 10–14.6% of total precipitation, and the runoff ranged between 7.3% and 10.5%. The rainwater evaporation results matched with the reported 76% for the Mexico Basin [20]. The rainwater infiltration and the irrigation water returns add into the natural groundwater recharges. The Tula and El Salado River sub-basins encompass the most extensive irrigation areas, and their water returns collect 35% of the total recharge. Surface water included the rainwater runoff, the local sub-watersheds sewage, and the imported wastewater from Mexico Megacity (main contribution, 1536 Mm$^3$). The baseline depicts unsustainable agriculture with untreated wastewater flood irrigation.

Water demand and outflows. The surface outflows were the evapotranspiration and the waters conveying to the Zimapan dam. The irrigation districts are the primary internal consumers, fulfilled by domestic sewage and the imported wastewater (59%, 831 Mm$^3$). Watersheds imported water consumptions were Tula, 36%; Actopan, 9%; El Salado, 8%; and Alfajayucan, 6%. Figure 5 shows the balance results. The surface outflow pours downstream into the Zimapan hydroelectric dam (839 Mm$^3$). The result is comparable to the annual concessioned volume (851.2 Mm$^3$) for electric generation in 2005 [22]. The slight difference (1.4%) might be due to uncertainty in the evapotranspiration calculation.

El Salado sub-basin groundwater presented strong overexploitation (−60 Mm$^3$), and its withdrawal was three times higher than the groundwater recharge because of the effect of the industrial sector and population. The Tepeji-El Salto aquifer was in equilibrium, despite the Tula-Tepeji manufacturing region, whose population and industries withdraw 95% of the total groundwater recharge. The remainder aquifers presented sustainable management with a low withdrawal/recharge ratio: 10% Alfajayucan, 27% Tlautla, 43% Tula, and 53% Actopan (Figure 5).

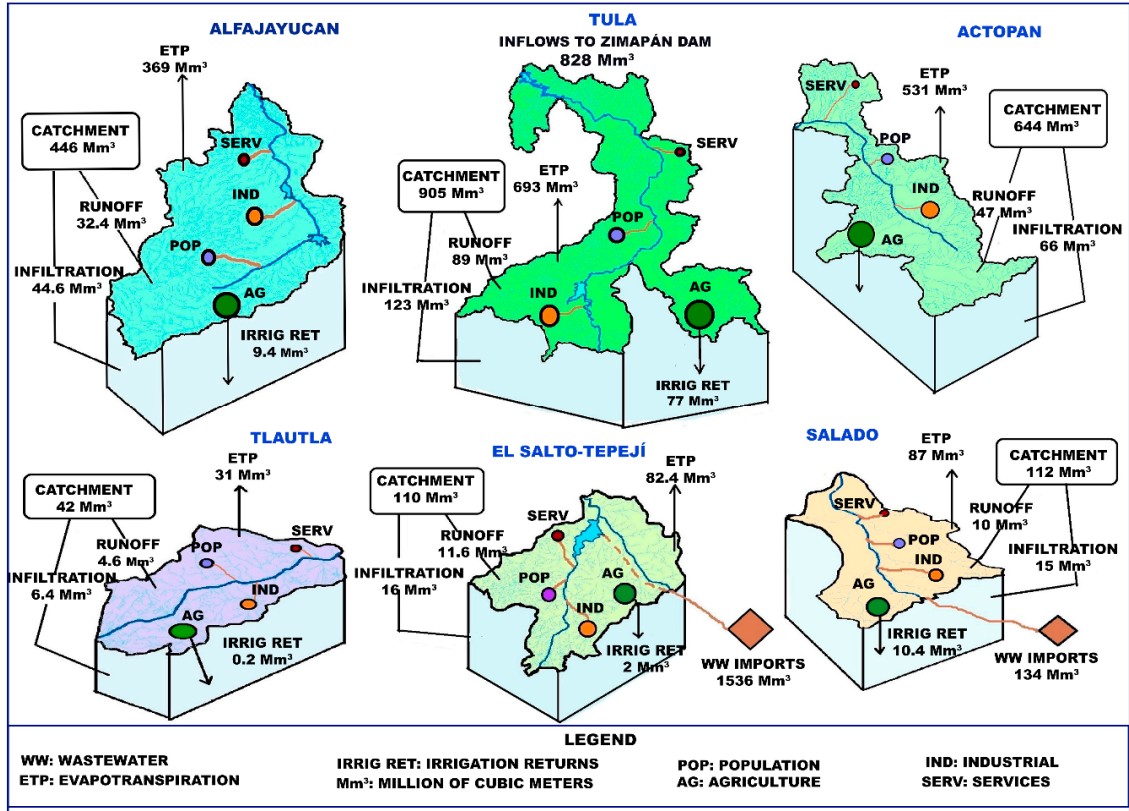

**Figure 5.** Calculated results in the sub-basins of Mezquital Valley in the baseline year 2005.

Freshwater availabilities. The rain infiltration was 473 Mm$^3$ for 711,450 people in the Mezquital Valley. El Salto-Tepeji and El Salado sub-watersheds presented absolute scarcity (<500 m$^3$/pers/y), based on the Falkenmark indicator. The Tula and Actopan River sub-basins were under stress conditions (<1700 m$^3$/pers/y) while the other sub-basins were without stress as they are sparsely populated. The average freshwater availability for the Mezquital Valley was 952 m$^3$/pers/y, under stress conditions.

The results lead to the following considerations:

- A sub-basin analysis is needed for management policies as their characteristics are not the same.
- Freshwater caption and the prevention of its contamination should be underlined in special programs for sub-basins El Salto-Tepeji and El Salado.
- A total of 59% of the imported untreated wastewater supports the agriculture irrigation of four sub-basins. Therefore, soil recuperating programs and remediation technologies should be designed, with particular emphasis on the Tula sub-basin.
- There is a misleading thought that rain is the first water input and support of valley agriculture because of the unsuitable evaluation of evapotranspiration.
- The Actopan and Alfajayucan sub-basins will not be benefit with the WWTP. Therefore, they need different local policy plans for water remediation.

*3.3. Transient Conditions Results*

3.3.1. Scenario 1. Steady-State (Reference)

The scenario considers the same unsustainable 2005 conditions, worsened by the inertial growth in the industrial, residential, and agriculture sectors. Although it includes the WWTP startup in Atotonilco de Tula, it does not affect the quantitative results but allows for a gradual improvement in the surface water quality for agriculture irrigation since 2017. Figure 5 shows the trends of demands,

inflows, and outflows for superficial and groundwater. Table 6 depicts the calculated values for the years 2030 and 2050.

**Table 6.** Results of transient scenarios. Water inflows and outflows in the Mezquital Valley ($Mm^3/y$).

| # | Name | Description | Source | 2017 | | 2030 | | 2050 | |
|---|------|-------------|--------|------|------|------|------|------|------|
| | | | | In-Flow | Out-Flow | In-Flow | Out-Flow | In-Flow | Out-Flow |
| 1 | Steady-state | Population, industrial, and irrigation growth rates lead the water demands | Groundwater | 391.6 | 267 | 408.6 | 313 | 431 | 403 |
| | | | Surface water | 1925 | 1294 | 1931 | 1524 | 1944 | **1966** |
| 2 | Reference + climate change effect | Rain infiltration and runoff gradually decrease 6.5% in 2050 | Groundwater | 371 | 267 | 363 | 313 | 353 | **403** |
| | | | Surface water | 1914 | 1294 | 1908 | 1524 | 1900 | **1966** |
| 3 | Wastewater splits to other state + new irrigation technologies | Wastewater inflow falls 31% in 2020 and irrigation demand gradually decreases 45% in 2050 | Groundwater | 341 | 267 | 350 | 310 | 350 | **370** |
| | | | Surface water | 1670 | 1294 | 1338 | 830 | 1338 | 667 |

In bold: The water demands (outflows) surpass the availability (inflows: water inlets + groundwater recharge).

Groundwater. Inertial demands will grow by 37% in 2030 and 76.5% in 2050. The population and agriculture needs will increase by 52% while those of industry, the most demanding sector, will rise by 60% (Figure 6a). In 2050, El Salado and Tula sub-basins would be the most water-consuming in the valley (38% and 35%, respectively) as they are the most populated and industrially developed, followed by Actopan and El Salto (15% and 8%, respectively). The four sub-basins constitute 96% of the valley demands. The Alfajayucan and Tlautla River groundwater use is low (<5%) due to the smaller population. Figure 6b shows El Salado sub-basin with the greatest overexploited groundwater since 2017, a condition that worsened in subsequent years. Impaired water management in the Mezquital Valley endangers the aquifers and urgent plans should guarantee the sustainable use of groundwater resources.

Surface water. Agriculture irrigation is the most demanding in the Valley. Figure 6c shows the growth in the demand of the sector from 2005 to 76% in 2050. The irrigation districts in the Tula and Actopan sub-watersheds will consume 72% of total surface water demand. In contrast, the Alfajayucan district consumption is merely 12%. The expansion of irrigation areas would increase the imported wastewater demand by 75% in 2030 and 100% in 2050, reducing the Tula River outflows, so that the Mezquital Valley would have deficit circumstances (see Figure 6d).

The inertial scenario demonstrates the need for action programs to reach sustainability and rational use of the water since the demands would be unmet.

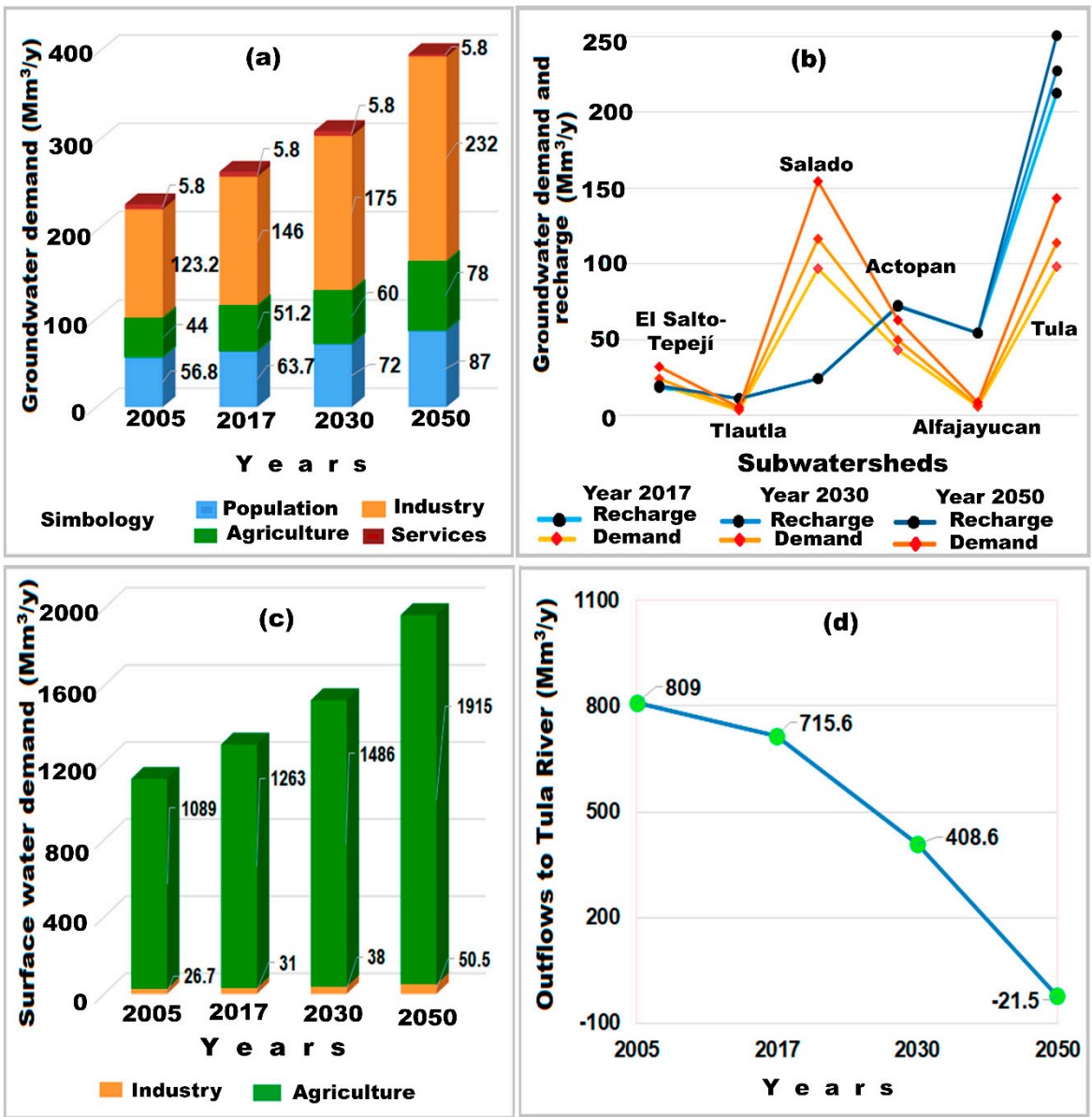

**Figure 6.** Scenario 1 (steady-state) main results. Calculated results in the Mezquital Valley. Values in Mm³/y. (**a**) Sectorial groundwater demands with industry relevance. (**b**) Yearly groundwater recharges vs. demands per sub-basin, where El Salado presents the highest hydric stress. (**c**) Sectorial water surface demands dominated by agriculture. (**d**) Tula River calculated outflow shows a sharp decline after 2017.

### 3.3.2. Scenario 2. Climate Change Perturbations

The scenario considers the same conditions as Scenario 1, worsened by climate change overcast. The analysis was focused on the natural freshwater availability impacted by the predicted local climate conditions in the Mezquital Valley, a 2 °C temperature rise, and a 6.5% precipitation drop in 2050 [66]. Table 1 shows the calculated inflows and outflows for each year. El Salto, Tula, El Salado, and Actopan sub-basins show the highest water scarcity. The freshwater availability decreases along the timeline, and the Alfajayucan and Tlautla sub-basins turn to hydric stress after an availability decay in 2030 (see Figure 7).

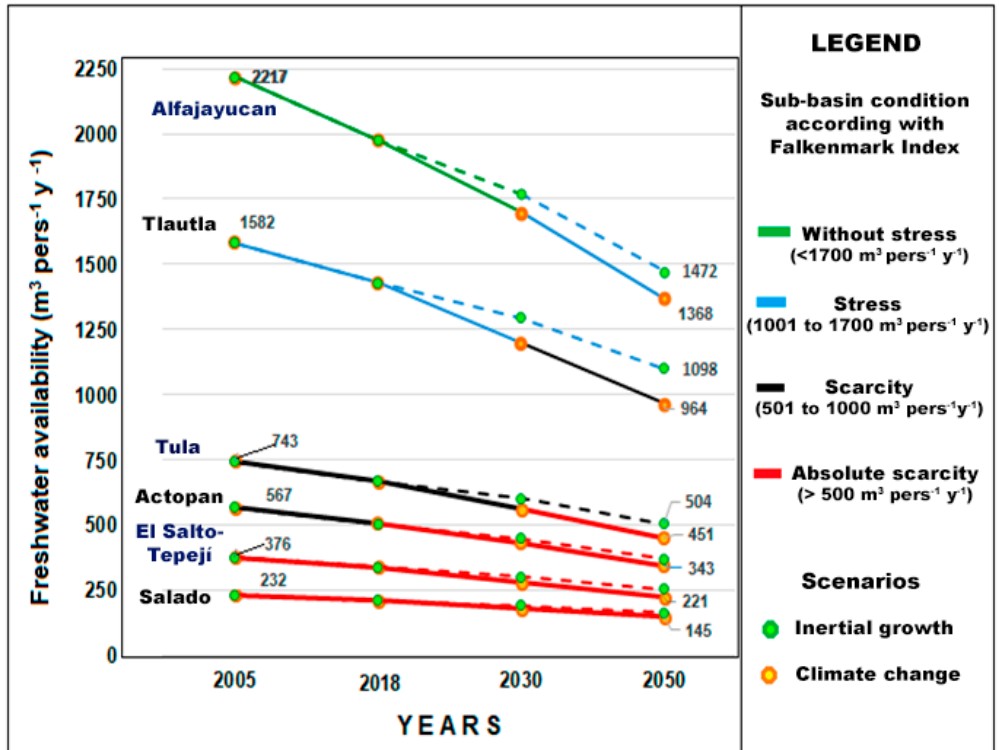

**Figure 7.** Freshwater availabilities in the Mezquital Valley sub-basins in the 2005–2050 period. Dashed lines only represent inertial growth rates (Scenario 1). Continuous lines consider inertial growth rates plus climate change conditions (Scenario 2).

Groundwater. The annual rainwater in the region would fall from 2290 Mm$^3$/y in 2005 to 2140 Mm$^3$/y in 2050 and, consequently, runoff and infiltration reduction would be reduced by 10% in 2050. Although evapotranspiration would rise 1% due to a higher temperature, its values decrease because of the precipitation decay. The runoff and infiltration reduce the groundwater recharge, the surface water inlets and, consequently, the water supply and the outflow.

Surface water. The calculated surface volumes for agriculture irrigation surpass the potential supply, as in Scenario 1. The influence of climate change in the surface water would not be so significant as long as the sewage importation continues. Generally, semidesert and desert zones bias climate change effects, due to the already existing hydric stress, such as the scenario results [21].

### 3.3.3. Scenario 3. Imported Wastewater Reduction and Adaptation Action, Sprinkler and Drip Irrigation

Scenario #3 is the most realistic. It considers the same conditions as the previous scenario, including the scheduled actions planned by the Federal government, and the 593 and 606 Mm$^3$/y reductions in imported wastewater from Mexico City. The volumes split towards new WWTP in Mexico State from 2020 with aggravating lack of water. As in China, the centralized water management integrates the wastewater claims into a water resource allocation agreement by the Federal government [69].

The implementation of two irrigation strategies (sprinkling and drip) was calculated, allowing for water savings. Table 6 displays the calculated inflows and outflows for 2030 and 2050. Figure 8 shows the irrigation reductions in the surface water demand (sprinkler 227 Mm$^3$/y and drip 473 Mm$^3$/y). Drip irrigation resulted more water-saving than the sprinkler, reducing 42% of the demand.

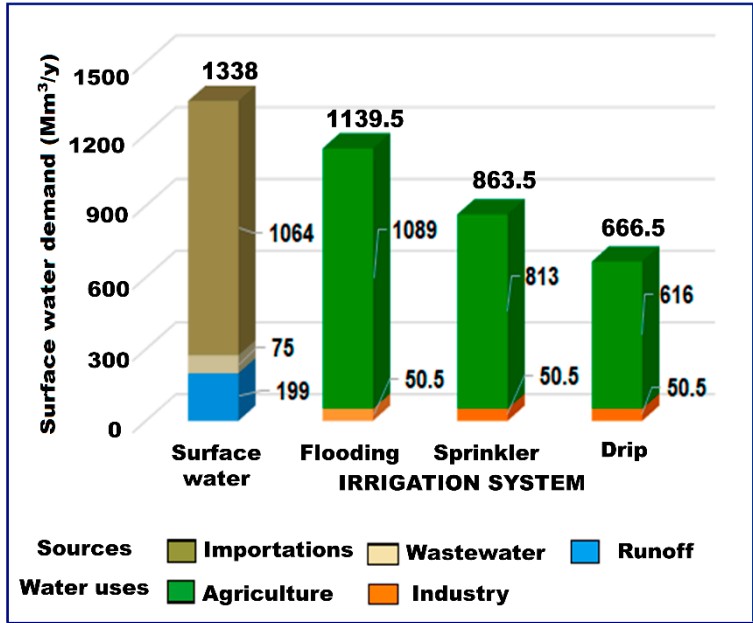

**Figure 8.** Scenario 3 calculated results for the year 2050. Wastewater importation from Mexico City is reduced by 660 Mm$^3$/y, and adaptation actions are the irrigation alternatives in the Mezquital Valley. Importation is the primary source of surface water and the domestic wastewater is near 1%. Drip irrigation appears to be the most efficient.

Drip irrigation might fulfill the surface water demand for agricultural activities in 2030, but the volume delivered to the Zimapan hydroelectric dam would not. Given that it would be not enough, decisionmakers should program sustainable intensive agricultural practices.

*3.4. Mezquital Valley Hydrological Balance Scenarios. Flow Diagrams*

Figure 9 shows the hydrological balance flows in Sankey diagrams for the baseline and the 2050 prospective scenarios.

Baseline. In 2005 (Figure 9a), water supplies and untreated wastewater importations were sufficient to fulfill the volume allotted to the Zimapan hydroelectric dam (606 Mm$^3$). Agriculture irrigation is the primary internal flow and evapotranspiration is the most significant outflow. Precipitation does not fulfill the surface water demands, so the wastewater importation guarantees the agriculture development in unsustainable conditions. The groundwater balance computes a net 149 Mm$^3$/y recharge.

Steady State, 2050. The hydrological balance (Figure 9b) predicts a lack of surface water due to the inertial growth. Agriculture irrigation demands 98.5% of the total surface water and the outflow to the Zimapan dam might fade away with a −21.5 Mm$^3$ deficit, jeopardizing the hydroelectric production. The groundwater balance is almost in equilibrium, but the lack of surface water results in an overexploitation risk.

Climate change, 2050. The balance (Figure 9c) forecasts a precipitation drop, lowering the rainwater inflow and depleting 7% of the water supply. Therefore, agriculture would consume 99.5% of the surface water and total lack of supply for hydroelectric production (outflow deficit: −40.5 Mm$^3$). The groundwater balance results in a mild overexploitation of −17 Mm$^3$/y.

Drip irrigation, 2050. It includes a wastewater inflow reduction due to a 36% withdrawal from new treatment plants in Mexico State, worsening the scenario. The balance (Figure 9d) includes drip irrigation to lower agriculture needs, the most water-demanding sector, but the water demand for Zimapan Hydroelectric Power Plant (989.2 Mm$^3$) [70] would not be fulfilled (177 Mm$^3$). The

surface impairment jeopardizes the groundwater sustainability (overexploitation of $-12$ Mm$^3$), and the scenario alerts policymakers to focus on intensive agriculture.

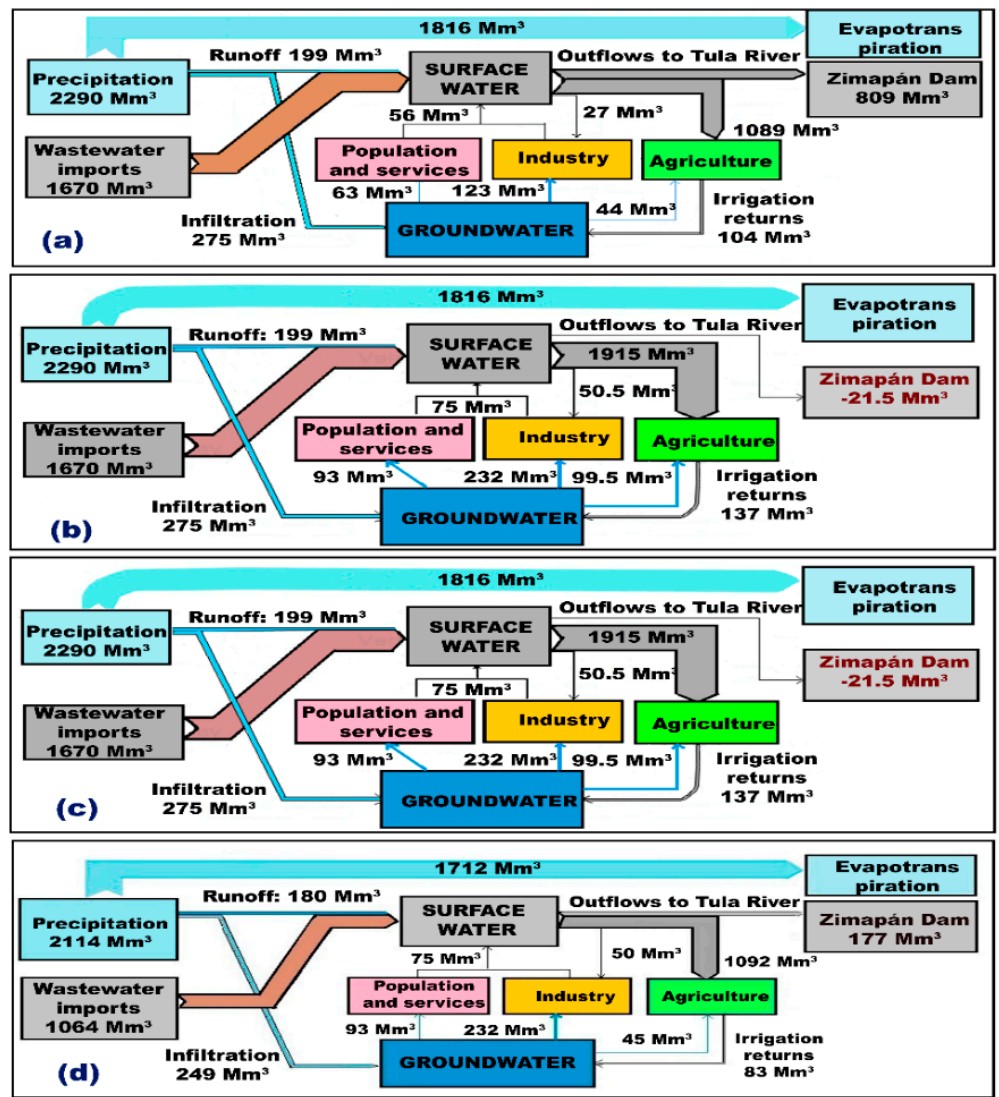

**Figure 9.** Flux diagrams for hydrological balance scenarios in the Mezquital Valley. Surface and groundwater annual fluxes (Mm$^3$/y). (**a**) Baseline, 2005. (**b**) Scenario 1: Steady-state reference, 2050. (**c**) Scenario 2: Climate change, 2050. (**d**) Scenario 3: Climate effects and drip irrigation, 2050.

## 4. Conclusions

The article illustrated the environmental damage caused by wastewater from big cities to the surrounding areas when the sewage is managed under unsustainable conditions.

This work studied a basin next to Mexico City as a case study. It calculated the water balance in simulated outcome scenarios, separating waste and freshwater use, as well as disposal for the industrial, agriculture irrigation, and population/services sectors. The conceptual model disaggregated the valley in sub-basins, which were calculated by the WEAP model, with suitable calibration, despite the complicated outlook of the basin. Flow diagrams were depicted for the 2005 baseline and future scenarios, considering internal returns and external inflows and outflows. The results were revealing and warned about the urgent need for efficient irrigation, modern agriculture techniques, and the internal recycling of industrial water.

A recognized paradox was demonstrated and quantified. The principal source of surface water is the wastewater importation from Mexico Megacity and not the runoff because evapotranspiration

represents 79.2% of the total precipitation. Rising demands of surface water would cause insufficient supply for the hydroelectric power plant requirements in 2030 and beyond. Agriculture irrigation is the most demanding sector, consuming as much as 58% of sewage waters in 2005, while in 2050 the volume would not be enough despite the introduction of efficient irrigation technologies.

Climate change will increase the hydric stress, but the effect on surface water is negligible. In contrast, the Wastewater Treatment Plants in Mexico State subtraction of imported surface water worsens its availability. Moderate groundwater overexploitation is predicted in 2050 ($-12.5$ to $-38.5$ Mm$^3$/y), but the forecast surface water deficit would be a risk to the deep aquifers, leading to more intense overexploitation.

The unsustainable conditions and lacking policy practices have caused environmental damage for more than 100 years. Although adaptation programs are designed for water management and ecosystem restoration, this research alerts of future unsolved demands based on predictive scenarios of water balance modeling. This work contributes to the knowledge of the environmental damage in the megacity's surroundings due to unsustainable sewage management.

**Author Contributions:** Conceptualization and project administration, E.O.-S., A.G.-M., C.G.-R.; methodology, software, validation, data curation, investigation, and supervision, S.C.-C., J.S.-S. and E.O.-S.; formal analysis, A.G.-M. and H.M.-N.; writing—original draft preparation, writing—review and editing, E.O.-S., S.C.-C., J.S.-S. and H.M.-N. All authors have read and agreed to the published version of the manuscript.

**Funding:** This research received no external funding

**Acknowledgments:** S.C.-C. thanks CONACyT for a doctoral scholarship. The authors recognize Hidalgo State Autonomous University for logistic support.

**Conflicts of Interest:** The authors declare no conflict of interest.

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
