# Peer review of "Megacity Wastewater Poured into A Nearby Basin: Looking for Sustainable Scenarios in A Case Study"

_water, doi:10.3390/w12030824_

Round 1

Reviewer 1 Report

This is a very interesting study on the adverse environmental effects of releasing untreated wastewater, focusing on Mexico city as the case study. Below are some minor comments, most of which are copyediting corrections. I suggest publication after minor revisions.

Manuscript language needs some improvements. I suggest proofreading by an expert. WEAP abbreviation should only be spelled out once in the first use. Please remove the full forms after you introduced its abbreviated form. Is there any justification for considering the same water use for individual crops? Page 5, line 159. Lines 179. 182: punctuation after water supply sources and cover vegetation is not appropriate. Please change the style. The same for lines 209, 240, 249, 254 and others. Please find all and correct. I suggest moving Tables 1S, 2S, 3S to the main manuscript as they contain important model inputs. Line 258, is "330% higher” right? Figure 5 resolution is very low, making all four images unreadable. Please provide high-resolution images. Please add the figure showing results from scenario 2 to the main manuscript Line 375 starts with “d)”. Please correct the sentence.

Reviewer 2 Report

This paper studies the water balance progression of realistic scenarios from 2005 to 2050 in the Mezquital Valley by using a WEAP model to calculate the water demand and supply. In addition, some significant suggestions were also obtained after discussing an illustrated case by authors. I found the idea interesting and valuable, although some major concerns arise when reading the document:

The introduction is loosely organized. The paragraphing is inappropriate which leads to the lack of coherence in contents and key points. What’s more, authors introduce the achievements which have been got in detail, but don’t give extension that can support authors’ own methods and innovations.

The motivation of considering different scenarios in the introduction part should be elaborated. Why these scenarios? Please give some more explanation.

Some assumptions were listed before modelling. The authors need explain more about the valley, then the readers can understand that these assumptions fit the reality. For example, why the valley can be considered as a set of six adjacent sub-basins with different size? Authors also should list more references about the assumptions for capacity in the third assumption.

The details of Figure 5 in Transient conditions results part are not clear enough. The author should provide high-quality figure and indicate the resource to make the paper more rigorous.

In Section 3.2 authors pointed out that the difference between surface outflow and the annual concessioned volume for electric generation is caused by losses in evaporation calculation. I think authors should give detail reason.

In addition, please check the typos, mathematical formulas and reference format.

Round 2

Reviewer 2 Report

the quality of the paper is well